# Flavor Differences of Edible Parts of Grass Carp between Jingpo Lake and Commercial Market

**DOI:** 10.3390/foods11172594

**Published:** 2022-08-26

**Authors:** Hongsheng Chen, Deyin Pan, Hongzhen Du, Jinming Ma, Baohua Kong, Jingjing Diao

**Affiliations:** 1College of Food Science, Heilongjiang Bayi Agricultural University, Daqing 163319, China; 2China-Canada Cooperation Agri-Food Research Center of Heilongjiang Province, Daqing 163319, China; 3College of Food Science, Northeast Agricultural University, Harbin 150030, China; 4National Coarse Cereals Engineering Research Center, Heilongjiang Bayi Agricultural University, Daqing 163319, China

**Keywords:** grass carp, flavor characteristic, volatile compound, Jingpo Lake, electronic nose, electronic tongue

## Abstract

This study investigated the flavor differences among three individual parts (abdomen, back, and tail) of Jingpo Lake grass carp (JPGC) and commercial grass carp (CGC). The growing environment and fish parts influenced the volatile compounds of the fish. The highest total contents of alcohols and ethers were found in the back of JPGC (*p* < 0.05). The combination of an electronic tongue and electronic nose (E-nose) could effectively distinguish the flavor differences between the different parts of JPGC and CGC by principal component analysis. Both the content of total free amino acids (FAAs) and content of amino acids contributing to the sweet and fresh flavors were higher in JPGC than CGC (*p* < 0.05). Among the ATP-associated products, the inosine 5’-monophosphate (IMP) contents of the back and tail of JPGC were higher (*p* < 0.05), but the abdomen content was lower (*p* > 0.05) than the respective contents in the corresponding parts of CGC. Sensory evaluation shows that JPGC had a better texture, odor, and taste, compared to CGC. Correlation analysis showed that the E-nose data and FAAs were highly correlated with the content of alcohols, aldehydes, and ethers. This study showed that the flavors of the different parts of JPGC differed significantly from those of CGC.

## 1. Introduction

Grass carp (*Ctenopharyngodon idellus*), typically a freshwater fish, is an important economic freshwater fish in China. In 2020, global carp production was 550 million tonnes, with 21% of the total freshwater aquaculture produced by China [1]. The main edible part of the fish meat provides high nutrient content and is a major contributor to the flavor of grass carp. Generally, the commercial grass carp (CGC) is bred in a high-density culture mode, and microorganisms in the water environment could easily cause the accumulation of undesirable flavors in the fish, thereby affecting consumers’ acceptance of cultured grass carp [2]. However, the meat quality of aquatic products, including grass carp, is affected by the different anatomical parts and region of origin, besides the growing environment [3,4]. Nowadays, wild grass carp is favored by consumers for its excellent sensory characteristics, such as its unique flavor and delicious taste. Jingpo Lake is a lava-dammed lake in China that supports a high abundance of wild grass carp. The lake is a national AAAAA-level tourist attraction in China, popular for sightseeing, and its annual winter fishing festival provides opportunities to sample some authentic local fish dishes. In recent years, wild grass carp from Jingpo Lake (JPGC) has been considered a local specialty. However, few studies have focused on the flavor differences between JPGC and CGC.

Flavor, including taste and odors, is an important quality characteristic in evaluating aquatic products. Some previous studies have reported methods for evaluating the non-volatile taste-active compounds in aquatic products, such as free amino acids (FAAs) and nucleotides, and odor-active volatile organic compounds (VOCs), such as alcohols, aldehydes, acids, ketones, esters, and hydrocarbons [5,6,7,8]. Moreover, the human olfactory and taste sensory systems are simulated by the electronic nose (E-nose) and electronic tongue (E-tongue) mechanisms to distinguish aromatic, hydrogen, broad-alcohol, sourness, bitterness, and saltiness, although these electronic sensors cannot detect specific compounds. By contrast, gas chromatography–mass spectrometry (GC-MS) can be used to analyze the specific substances but cannot identify specific tastes and odors. Therefore, several studies used an E-nose to detect the overall odor of foods, solid-phase microextraction (SPME) coupled with GC-MS to analyze the specific compounds [7], E-tongue to detect the overall taste of food, high-performance liquid chromatography to determine the FAAs, and ATP-associated compounds to identify individual taste compounds [9]. 

Numerous studies have outlined various differences in sensory value between farmed and wild fish, and farmed fish are generally considered less tasty [10,11,12,13]. Hu et al. [14] characterized the difference in nutritional value and flavor among bighead carp raised in the cold-water reservoir (XHK), natural lake (PY), and common culture pond (NC) and found that the strongest fishy odor was detected in cultured walleye; the pleasant plant-like odor was detected in natural lakes and cold-water reservoirs walleye. Jia et al. [15] found that the volatile compounds of largemouth bass cultured in an aquaculture system using a land-based container with recycling water imparted a strong, intense fruity flavor, whereas largemouth bass cultured in a traditional pond system had a strong and pungent flavor. Wang et al. [16] clarified the physical characteristics and volatile matter of muscle of the Yellow River carp (*Cyprinus carpio haematopterus*) grown in the wild, as well as the different farming systems, and found that the best muscle quality and flavor of the Yellow River carp were grown in the wild. However, few studies have examined the flavor differences in different edible parts of grass carp between Jingpo Lake and commercial market. Hence, the flavor characteristics of different edible parts of farmed and wild grass carp assessments are essential for informing the consumer about the true sensory variation, based on the facilitating knowledge, rather than belief-based choices.

Therefore, this study aimed to analyze the flavor differences in the abdominal, back, and tail muscles of JPGC (JA, JB, and JT, respectively; Figure 1a) and CGC (CA, CB, and CT, respectively; Figure 1b) by using an E-nose, HS-SPME-GC-MS, E-tongue, FAAS, ATP-associated compounds, and sensory analysis. The results obtained via these techniques are conducive to a better understanding of the flavor differences in edible parts of grass carp between Jingpo Lake and the commercial market and provide a theoretical basis regarding the use of Jingpo Lake regional grass carp varieties.

## 2. Materials and Methods

### 2.1. Animals and Ethics Approval

All methods used in this study complied with the Chinese National Guidelines for the use and care of laboratory animals. The animal experiment protocol was approved by the Science and Technology Ethics Committee of Heilongjiang Bayi Agricultural University (SPXY2022006).

### 2.2. Materials

Amino acid standards were purchased from Waters Corp. (Milford, MA, USA). Nucleotide standards were purchased from Shanghai Yuanye Biotechnology Co. (Shanghai, China). HPLC-grade acetonitrile and methanol were acquired from Fisher Scientific (Fair Lawn, NJ, USA).

### 2.3. Sample Preparation

Eighteen fresh grass carp samples (2500 ± 150 g, 42.0 ± 4.5 cm in length) were purchased from a local supermarket (Daqing, China). Another 18 fresh grass carp samples (2500 ± 150 g, 44.0 ± 3.7 cm in length) were legally obtained from Jingpo Lake (Ningan, China) by local fishermen using long lines or nets, respectively. All samples were kept on ice and transported to the laboratory. In the laboratory, the head, guts, bones, scales, and skin were removed, and the meat of the grass carp was collected. The collection of the meat cuts (red area) from JPGC and CGC are shown in Figure 2. Then, the two types of grass carp were each divided into three groups. Each group of grass carps (*n* = 6) was treated as a batch and independently processed. Three different cuts of meat were collected from both sides of JPGC (JA, JB, and JT) and CGC (CA, CB, and CT), and each cut of meat was individually vacuum packaged and stored at −80 °C until use (within 7 days) [17].

### 2.4. Physicochemical Analysis

Crude protein content was determined using the micro-Kjeldahl method (GB 5009.5–2016). Crude fat content was determined using the Soxhlet extraction method (GB 5009.6–2016). Moisture content was determined using the 105 °C direct-drying method (GB 5009.3–2016). Ash content was determined using the muffle furnace volatilisation constant weight method (GB 5009.4–2016).

### 2.5. Volatile Compound Analysis

The volatile compounds were extracted from the samples using headspace solid-phase microextraction (HS-SPME) and analyzed using a GC-MS system (GCMS-QP2020 NX, Shimadzu Corp., Kyoto, Japan), based on the method of Du et al. [18]. Briefly, 3.0 g minced grass carp meat was put into 20 mL headspace sample vials. After mixing with 4 μL ortho-dichlorobenzene, SPME fiber coated with polydimethylsilocane/divinylbenzene/carboxen (PDMS/DVB/CAR) (50/30 μm, Supelco, Bellefone, PA, USA) was inserted into headspace sample vials and exposed to the vial headspace at 45 °C for 45 min. Cap WaX capillary columns (60 m × 0.25 mm × 0.25 μm) were used to separate the volatile compounds. The carrier gas was helium with a flow rate of 1 mL/min. The chromatograph oven temperature was held at 40 °C for 4 min, raised from 40 °C to 135 °C at 3 °C/min, raised from 135 °C to 200 °C at 5 °C/min, and subsequently raised from 200 °C to 230 °C at 15 °C/min, with a final holding time of 5 min. Volatile compounds were identified by comparing the experimental mass spectra with a mass spectra library from NIST17. The final volatile compound contents were calculated based on the target compound’s peak area divided by the internal standard’s peak area (expressed as μg/kg).

### 2.6. E-Nose Analysis

The odorant characteristics of different parts of grass carp were analyzed by an E-nose system (PEN3 Airsense, Schwerin, Germany) [19]. The sensor probes of W1C, W5S, W3C, W6S, W5C, W1S, W1W, W2S, W2W, and W3S were installed in the E-nose system. They are highly sensitive to the aromatic constituents and benzene (W1C), nitrogen oxides (W5S), aroma and ammonia (W3C), hydrides (W6S), short-chain alkane aromatic component (W5C), methyl compounds (W1S), sulfides (W1W), alcohols, aldehydes, ketones (W2S), organic sulfides (W2W), and long-chain alkanes (W3S).

### 2.7. E-Tongue Analysis

E-tongue samples were prepared as reported by Zhang et al. [20]. The taste sensing system (TS-5000Z, Insent, Inc., Atsugi-Shi, Japan) was composed of eight sensors: bitterness, umami, aftertaste-A (aftertaste-astringency), saltiness, richness, sourness, aftertaste-B (aftertaste-bitterness), and astringency.

### 2.8. Determination of FAAs

The FAAs were determined according to the assay reported by Li et al. [21], with slight modifications. Briefly, 0.5 mL of the mixed standard solution (17 amino acids + NH_3_: the concentration of Cys is 0.25 mmol/L, and the concentration of the other compounds (0.5 mmol/L) was added to a 5 mL glass-stoppered graduated test tube, followed by the addition of 0.5 mL of 0.1 mol/L Na_2_B_4_O_7_ aqueous solution and 1% 2,4-Dinitrofluorobenzene (DNFB) acetonitrile solution. The mixture was shaken well and reacted at 60 °C for 1 h in the dark. After the reaction was completed, the test tube was placed in cold water to cool. The volume was adjusted to 5 mL with 0.02 mol/L Na_2_HPO_4_–0.02 mol/L NaH_2_PO_4_ aqueous solution. The solution was mixed, and the absorbance was measured in a spectrophotometer.

### 2.9. ATP-Associated Compounds and K-Values

Extraction and HPLC analysis of ATP-associated compounds were performed according to the method of Zhuang et al. [22], with some modifications. Briefly, 4.0 g of the sample was homogenized in a 50 mL centrifuge tube, followed by the addition of 10 mL of HClO_4_ with a mass fraction of 10%, then homogenized for 2 min and centrifuged at 10,000 rpm for 15 min. The supernatant was filtered, and the pellet was washed with 5 mL of 5% HClO_4_. After a second centrifugation, the supernatants were pooled. The pH of the supernatant was adjusted to pH 6.5. The fluid was left to stand for 30 min and then quickly diluted to 50 mL, shaken well, passed through a 0.45 μm membrane, and measured. All operations were carried out at 4 °C. The *K*-value is equal to the percentage of the sum of hypoxanthine (Hx) and inosine (HxR), the two bitter nucleotides formed from the degradation of ATP to the total amount of ATP and other ATP-associated degradation products.

### 2.10. Sensory Analysis

The sensory evaluation was based on the method of Lazo et al. [23], with minor modifications. The trained team, composed of twelve members, conducted a sensory evaluation using individual light- and temperature-controlled booths to provide privacy and comfort. All samples were steamed at 100 °C for 15 min. The steamed samples were placed in petri dishes, coded, and made available to panelists at room temperature. Mineral water was used to cleanse the tastebuds of the panelists to ensure the accuracy of the evaluation. All tests were administered on the identical day. Each test was independent, and there were three tests. Texture characteristics (hardness, springiness, and chewiness), odor (fishy and earthy), and taste (fishy and earthy) descriptors of intensity were evaluated. Intensities of these features were rated using a 5-point scale, where 1 point corresponded to the lowest, with 5 points as the highest intensity. The sensory evaluation was repeated 3 times on each sample.

### 2.11. Statistical Analysis

Data analyses were performed using the Statistix 8.1 software package (Analytical Software, Inc., St. Paul, MN, USA), and the results were presented as mean ± standard error (SE). All graphs were drawn using Origin 2022 (OriginLab Corp., Northampton, MA, USA). The principal component analysis (PCA) was performed by SPSS 22.0 (Analytical Software, New York, NY, USA). One-way analysis of variance (ANOVA) with Tukey’s multiple comparison test was used to distinguish significant differences among samples (*p* < 0.05).

## 3. Results and Discussion

### 3.1. Physicochemical Parameters

The results of the physicochemical parameters are shown in Table 1. The crude protein and ash of JA, JB, and JT were higher than those of CA, CB, and CT (*p* < 0.05), and there were no significant difference between JA and JT (*p* > 0.05). The crude fat of JA, JB, and JT were lower than those of CA, CB, and CT (*p* < 0.05). This may be attributed to the different conditions of movement and diets of grass carp in different growing environments, thus resulting in a high protein and ash and low fat of meat. This corresponds to the morphological differences (Figure 1) in JPGC and CGC. There were no significant differences in moisture contents between JPGC and CGC (*p* > 0.05).

### 3.2. Volatile Compounds

A total of 119 volatile compounds, including 53 esters, 29 alcohols, 14 hydrocarbons, 7 aldehydes, 8 ethers, 3 ketones, and 5 others, were identified in different parts (abdomen, back, and tail) of JPGC and CGC. A total of 83 volatile components were identified in JPGC, including 35 esters, 23 alcohols, 9 hydrocarbons, 6 aldehydes, 6 ethers, and 3 other compounds, and a total of 53 volatile components were found in CGC, including 23 esters, 12 alcohols, 7 hydrocarbons, 4 aldehydes, 3 ketones, 2 ethers, and 2 other compounds. As shown in Table 2, JB was the origin of most of the volatile compounds in JPGC.

The total alcohol content was significantly higher in JPGC than in CGC, and there were discrepancies between different parts of JPGC and CGC. In particular, JB had significantly more than the other two anatomical regions. Similar results were reported by Wang et al. [24], who identified 56 and 55 volatile compounds in the grass carp’s back and abdomen parts, respectively, with the main volatile compounds being alcohols, which were more abundant in the dorsum than in the abdomen. The content of 1-nonanol in CA and CB was higher. JA contained a small amount of this organic compound; none was detected in JB, and the difference between JA and CA was significant (*p* < 0.05). This straight-chain fatty acid alcohol has a dusty and greasy odor [25] and was previously found in mussels (*Mytilus galloprovincialis* Lmk.) [26]. Cai et al. [27] also found 1-nonanol when studying the flavor of triploid common carp. Compound (*R*)-2,4-dihydroxy-*N*-(3-hydroxypropyl)-3,3-dimethylbutyramide was found in JB only. It is the precursor of vitamin B5 and has a slight special smell. The alcohol most detected in the grass carp samples was 1-hexanol. It was found at greater concentrations in JA, CA, JB, and CB than in JT and CT (*p* < 0.05), and CB contained more than that found in JB, but there were no significant differences among JA, CA, JB, and CB (*p* > 0.05). It has a fruity and fragrant aroma. The lipid oxidation pathway contributes to the formation of linear alcohols, which may be why the 1-hexanol content of JPGC was lower than that in CGC. The 1-octanol content of JB was higher than that in CB (*p* < 0.05). This alcohol is commonly found in aquatic products and provides a dry, sweet, sharp, fatty, waxy aroma with citrus-, orange peel-, and rose-like aromas [28].

Aldehydes have a very low odor threshold and contribute significantly to the overall flavor level of foods [29]. The total aldehyde content of JT was significantly higher than that of the other parts (*p* < 0.05). The aldehydes detected in the grass carp samples mainly included 2,5-dihydroxybenzaldehyde, 1-nonanal, and 3-methylbutanal. The 1-nonanal contents of JA, JB, and JT were higher than those in CA, CB, and CT (*p* < 0.05), but there were no significant differences between JA and JB (*p* > 0.05). This aldehyde may be derived from the oxidation of unsaturated fatty acids [30] and can provide fatty, citrus, and green aromas [31]. It also emits a strong aromatic odor, which can mask other odors at low concentrations [32]. The 2,5-dihydroxybenzaldehyde contents of JT, CA, CB, and CT were higher than those in JA and JB (*p* < 0.05), and there were no significant differences among CA, CB, and CT (*p* > 0.05). This natural antimicrobial agent inhibits the growth of *Mycobacterium avium* subsp. The contents of 3-methylbutanal in JA, JB, and JT were higher than those in CA, CB, and CT (*p* < 0.05). It is a Strecker aldehyde derived from Leu and described as having peach-like, chocolate, and malty flavors [33]. It may have formed during the SPME process. Hexanal was detected in JT only. It has been reported to cause fishy or green leafy odors in aquatic products [34]. Vanillin has a vanilla aroma and has been reported to be the most potent aldehyde in aquatic products [35]. This aldehyde was detected in JA and JB in our study. Isovanillin was detected in both CB and JT, but with significantly higher levels in the latter (*p* < 0.05). As an aroma ingredient, it is quite distinct from vanillin. Its fragrance can change with a change in the ambient temperature. Among these detected aldehydes, 2-dodecenal had a promoting effect on the fishy smell, and it was present in JT at significantly lower concentrations than in CT.

The only three ketones detected in this study were xanthoxylin, 5-decanone, and 5,5-dichloro-4-spirohexanone. They were found exclusively in all three parts of CGC. Wang et al. [24] also detected relatively low levels and types of ketones in grass carp.

The total ester contents in JT, JB, and CB were significantly higher than those in JA, CA, and CT (*p* < 0.05), and there were no significant differences between JB and CB (*p* > 0.05). Esters are another class of organic compounds with unique odors, usually formed by the reaction of carboxylic acids and alcohols and responsible for the formation of the characteristic aroma of aquatic products, which can mask rancid odors but also provide positive floral and fruity aromas to products [36]. Of the 53 esters detected in our study, methyl esters were the main compounds. Nineteen methyl esters were successfully identified, including methyl myristoleate, methyl 2-hydroxystearate, dimethyl dl-malate, methyl acetylglycinate, hexacosanoic acid, methyl ester, methyl benzoate, (2-bromomethyl-[1,3]dioxolan-2-yl)-acetic acid, methyl ester, butanoic acid, 4-ethoxy-, methyl ester, (*E*)-3,7,11-trimethyldodec-2-enoic acid, methyl ester, heptadecanoic acid, 9-methyl-, methyl ester, pentanoic acid, 2-hydroxy-3-methyl-, methyl ester, carbamodithioic acid, diethyl-, methyl ester, trimethylene borate, methyl octa-*O*-methyllaminaribionate, methyl-(aminosulfanyl)formate, tetraethyleneglycol monomethyl ether, methyl 3-hydroxytetradecanoate, methyl 5-methoxy-3-oxovalerate, 7-hexadecenoic acid, (*Z*)-methyl ester. Methyl esters impart fruity odors to meat products [37]. Methyl benzoate was the most abundant methyl ester in the grass carp samples in our study. The contents of methyl benzoate in JT, CT, and JB were higher than those in JA and CB (*p* < 0.05), although there were no significant differences between JB and CB (*p* > 0.05). Methyl benzoate provides prune, lettuce, herb, and sweet notes. Isopropyl palmitate (fatty odor) was detected exclusively in JA and JB. Ethyl 2-methylbutyrate, characterized as having an apple or apple-like aroma, was detected in JB. It contributes positively to the flavor of JPGC.

Although 15 hydrocarbons were identified, they had little effect on the overall flavor of grass carp because of their high odor threshold values [38]. CT had the highest total hydrocarbon level, followed by JT, which was consistent with the measurement results of the E-nose W1S sensor. The hydrocarbon most detected in the grass carp samples was heneicosane. The heneicosane contents in JT and CT were higher than those in JA, JB, CA, and CB (*p* < 0.05).

Among the eight ethers identified, six were found in JB, which had the highest total ether content and included estragole, anethole, 2-(2-methoxyethoxy)ethanol, 2-[2-(hexyloxy)ethoxy]-ethanol, tetraethylene glycol diethyl ether, ethanol, and 2-(2-ethoxyethoxy)-. The most abundant was estragole, characterized by an anise-like aroma, which contributed to the flavor of JPGC. Anethole was detected in all sample groups but was significantly highest in JT. It has a unique fennel smell and corresponding sweetness and can be used in all foods, especially pastry foods. A pleasant fragrance positively influences the flavor of grass carp.

Aside from the above compounds, some flavor compounds, such as 3-phenylindole, 1,2-bis(trimethylsilyl)benzene, butylated hydroxytoluene, azulene, and naphthalene, are challenging to classify. Naphthalene was detected in CB and CT. The substance has a strong tar odor and adversely affects the odor of CGC. These unpleasant odors may come from water or environmental pollution.

### 3.3. E-Nose

An E-nose is a device that can precisely distinguish odor differences between different samples by mimicking the structure of the human nose [39]. The spatial distribution and distances of different parts of JPGC and CGC aroma were analyzed by PCA, and the results are presented in Figure 3a. Based on the location and spatial distribution of data clusters (represented by each sample type), the six samples of grass carp were well-differentiated, indicating that they could be completely separated by E-nose. The first and second principal components (PC1 and PC2) explained 71.90% and 19.10% of the variance, respectively, and the cumulative contribution was 91.00%, which indicated that the main components were able to reflect all the features of volatile odors of JPGC and CGC in the different parts. The differences between parts of JPGC and CGC were mainly on PC1. Four groups (JA, CA, CB, and CT) were located on the positive side of PC1 and corresponded to the E-nose sensors of W6S, W3S, W1S, W2S, W1W, and W2W. The remaining groups (JB and JT) were located on the negative side of PC1 and corresponded to the W5S, W1C, W3C, and W5C sensors. Again, these results emphasize that the E-nose device can productively discriminate different parts of JPGC and CGC. The response values of the E-nose to the different parts of JPGC and CGC are shown in Figure 3b. There was a large difference between JT and CA in the W1S sensor, but only minor changes in the other nine sensors. This may be because raw fish meat has just a few overall volatile substances, thus resulting in little difference in the overall odor characterized by the E-nose.

### 3.4. E-Tongue

The E-tongue works by converting electrical signals into taste signals, and it has a low sensory threshold, which can eliminate the subjectivity of sensory evaluation [40]. The PCA results of the taste responses to different parts of JPGC and CGC are shown in Figure 4a. Of the total variance, 95.20% (>90%) was explained by PC1 and PC2, which accounted for 83.30% and 11.90% of the variance, respectively. All parts of CGC (CA, CB, and CT) were clustered in the second and third quadrant of PC1, which was related to aftertaste-A, sourness, and astringency. By contrast, all parts of JPGC (JA, JB, and JT) were clustered in the first and fourth quadrants, which were correlated with the other taste sensors. The proximity between JT and JB may result from their similar saltiness. The dataset for JA produced a cluster with different taste qualities from the other groups, having a unique combination of bitterness, umami, richness, and aftertaste-B. Simultaneously, JA was far from CA, which may be due to the difference in the bitterness and aftertaste-B of this part between the two fishes.

Figure 4b shows radar images of the taste results for the six groups of samples. It was evident that the umami and richness of JA were significantly higher than those of any other group, which is consistent with the results for the FAAs and ATP-related products (Section 3.4), thus indicating that JPGC has a more pleasant taste than CGC. There were no significant differences in saltiness, aftertaste-B, aftertaste-A, sourness, astringency, and bitterness flavors among all samples.

### 3.5. FAAs

FAAs play a major role in the flavor of aquatic products because of their taste and interaction with other flavor compounds. As shown in Table 3, 17 FAAs were detected, which is largely consistent with the results of Wang et al. [41] and Wu et al. [42]. The main FAAs in grass carp muscle are Glu, Asp, and Lys, accounting for more than 37% of the total FAAs in grass carp muscle. However, a low content of Glu and high contents of Ser and Pro have been reported in largemouth bass that were reared in a traditional pond system [15]. FAAs have various taste characteristics, such as umami, bitterness, and sweetness, and are important contributors to the formation of food flavor. As shown in Table 3, Thr, Ser, Gly, and Ala mainly contributed sweetness; Arg, His, Ile, Leu, Met, Phe, and Val contributed bitterness; and Asp and Glu mainly contributed to the umami taste. Glu can synergize with inosine 5’-monophosphate (IMP), which greatly enhances the umami taste of food [43]. Moreover, Glu was shown to serve as an energy source for the physiological regulation of the yellow drum (*Nibea albiflora*) exposed to cold and starvation stress [44].

JA had the highest levels of sweet amino acids. The contents of the 17 FAAs detected in JA were significantly higher (*p* < 0.05) than those in CA, except for Cys, Val, Ile, Try, and Pro, but their contents were still higher than those in CA. The contents of Asp, Glu, Gly, Ala, Leu, Phe, Lys, His, and Pro detected in JB were significantly higher (*p* < 0.05) than those in CB. The detected contents of Asp, Thr, Glu, Gly, Ala, Met, Leu, Phe, Lys, His, Arg, and Pro in JT were significantly higher than those in CT. Overall, the contents of FAAs in various parts of JPGC were significantly different from those found in CGC. Although the content of the bitter amino acids was also higher in JPGC than in CGC, IMP has an inhibitory effect on sourness and bitterness; thus, it can confer a pleasant and richer taste to the food.

### 3.6. ATP-Associated Compounds and K-Values

ATP-related compounds are widely used as indicators to assess the deterioration of aquatic products (*K*-value and its derived values). As shown in Table 4, the *K*-values of the six groups of samples were not significantly different, and the *K*-values were less than 20, thus indicating that all six groups of samples were of first-level freshness. The ATP contents of JA and JT were significantly higher (*p* < 0.05) than those in CA and CT, respectively, and the content in JB was higher than that in CB, albeit not significantly (*p* > 0.05). Among these ATP-related compounds, IMP, ADP, and AMP are responsible for the acceptable taste of umami, while Hx and HxR affect the freshness of fillets, imparting unpleasant off-flavor, bitterness, and an undesirable taste [45]. In addition, the ADP contents of JA and JB were significantly higher (*p* < 0.05) than those in CA and CB, respectively, and the content in JT was higher than that in CT, albeit not significantly (*p* > 0.05). The AMP contents in each of the three parts of JPGC were significantly higher (*p* < 0.05), compared to their respective parts in CGC. The IMP contents of JB and JT were higher than those in CB and CT, respectively, albeit the differences were not significant (*p* > 0.05). The HxR content of JA was lower than that in CA (*p* < 0.05). The Hx contents of JB and JT were lower than those in CB and CT, respectively (*p* < 0.05). To sum up, the overall taste of JPGC was superior to that of CGC, which may be closely related to the high quality of its growth environment. Although the IMP content of JPGC was lower or higher than that of CGC, the differences were not significant. There was a significant difference in Glu content between JPGC and CGC, and because IMP can synergistically react with glutamate to produce a strong umami taste, JPGC is superior to CGC, regarding the umami level. This result is consistent with that of the E-tongue PCA.

### 3.7. Sensory Evaluation

The results of the sensory evaluation are shown in Table 5. The tenderness, springiness, and chewiness of JA and JB were higher than those of CA and CB (*p* < 0.05), and there were no significant differences in tenderness and chewiness between JT and CT (*p* > 0.05). Drengstig and Johnston [46,47] also reported that low lipids and high protein also improved meat tenderness, springiness, and chewiness. In terms of odor, the earthy and fishy odor of each part of JPGC was significantly lower than that of each part of CGC (*p* < 0.05). Similar results were also reported by Farmer et al. [48], who found that the fishy flavor in farmed salmon was higher than that of the wild river salmon. In regard to taste, the score of earthy taste in JA, JB, and JT were significantly lower than those of CA, CB, and CT (*p* < 0.05). The score of fishy in CB was higher than that of JB, and there were neither significant differences between JA and CA, nor between JT and CT (*p* > 0.05). In summary, JPGC had better texture, odor, and taste, compared to CGC.

### 3.8. Correlation Analysis between E-Nose, FAAs, and GC–MS

Correlation analysis was used to provide further evidence for the relationship between E-nose, GC-MS, and FAAs. As shown in Figure 5a, the concentration of alcohols and hydrocarbons were positively related to the signal intensities of the W3C (*r* = 0.37 and 0.33), W2W (*r* = 0.43 and 0.48), and W5C (*r* = 0.47 and 0.61) sensors. The W1S, W1W, W2S, and W2W sensors were positively related to the concentration of aldehydes (*r* = 0.52, 0.60, 0.50, and 0.77, *p* < 0.05), and the W2W sensor had a positive correlation with esters (*r* = 0.82, *p* < 0.05). Shi et al. [19] and Zhang et al. [49] also found that the W1C, W3C, and W5C sensors were sensitive to hydrocarbons and alcohols. From the current results, the W1C and W5C sensors had negative correlations with aldehydes (*r* = −0.81 and −0.45, *p* < 0.05), and W5S was negatively associated with the ketones and esters (*r* = −0.50 and −0.46, *p* < 0.05). The above results indicated that the E-nose could distinguish between different parts of grass carp by the volatile compounds.

Figure 5b shows the correlation between FAAs and volatile flavor components. All FAAs showed a negative relation to the ketones and strong positive relation to the aldehydes. This may be because the source of aldehydes was associated with proteolytic phenomena and FAAs degradation. Cys and Met were positively related to the esters (*r* = 0.82 and 0.49, *p* < 0.05), and Ala, Phe, and His were positively related to the ethers (*r* = 0.42, 0.45, and 0.66, *p* < 0.05). Similarly, Merlo et al. [50] reported that the FAA content was correlated with volatile components. From the above results, it is plausible that the amino acid metabolism contributed to the formation of the volatile components.

## 4. Conclusions

JPGC and CGC differ in E-nose, HS-SPME-GC-MS, E-tongue, FAAS, ATP-associated compounds, and sensory analyses. The significant differences in the volatile compounds among different anatomical regions of JPGC and CGC were in the contents of alcohols, esters, and hydrocarbons. The highest total contents of alcohols, esters, and hydrocarbons were detected in JB. The umami, sweet amino acid content, and total FAAs of JA, JB, and JT were significantly higher than those in CA, CB, and CT. The highest content of IMP was detected in JB. The E-tongue, combined with the E-nose, could distinguish between differences in the taste and smell of each edible part of JPGC and CGC. Correlation analysis indicated that the FAAs are responsible for the formation of ketones and aldehydes. The JPGC had better texture, odor, and taste, compared with CGC. This study can provide useful information for elucidating the flavor characteristics of edible parts of JPGC.

## Figures and Tables

**Figure 1 foods-11-02594-f001:**
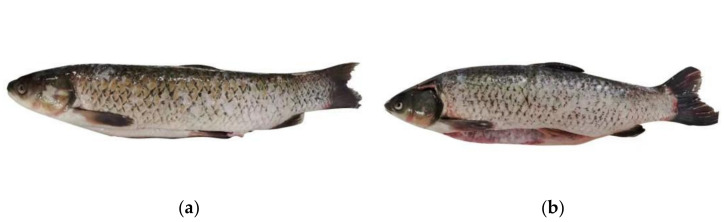
Comparative pictures of (**a**) grass carp from Jingpo Lake (JPGC) and (**b**) commercial grass carp (CGC).

**Figure 2 foods-11-02594-f002:**
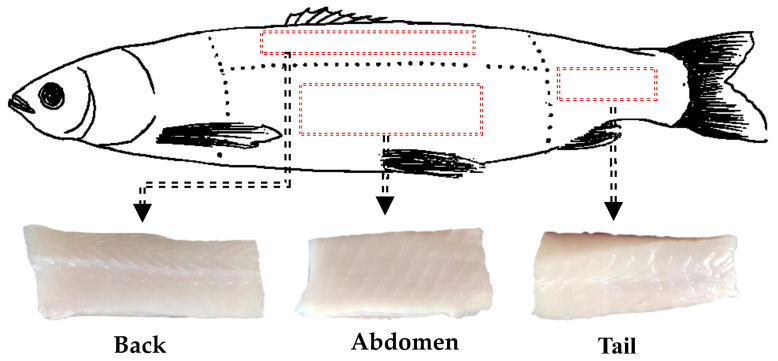
Schematic diagram of the collection of meat (red area) of different parts of grass carp.

**Figure 3 foods-11-02594-f003:**
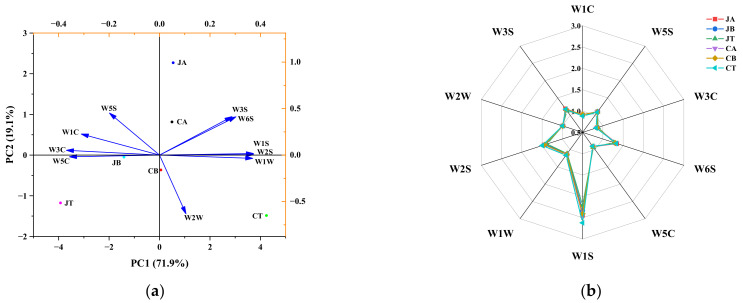
Principal component analysis (PCA) (**a**) and radar chart (**b**) of E-nose data for different parts of JPGC and CGC. JPGC: grass carp from Jingpo Lake; CGC: commercial grass carp; JA: abdomen of JPGC; JB: back of JPGC; JT: tail of JPGC; CA: abdomen of CGC; CB: back of CGC; CT: tail of CGC.

**Figure 4 foods-11-02594-f004:**
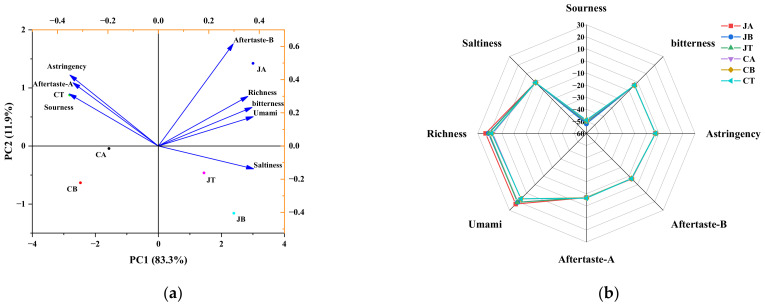
Principal component analysis (PCA) (**a**) and radar chart (**b**) of E-tongue data for different parts of JPGC and CGC. JPGC: grass carp from Jingpo Lake; CGC: commercial grass carp; JA: abdomen of JPGC; JB: back of JPGC; JT: tail of JPGC; CA: abdomen of CGC; CB: back of CGC; CT: tail of CGC.

**Figure 5 foods-11-02594-f005:**
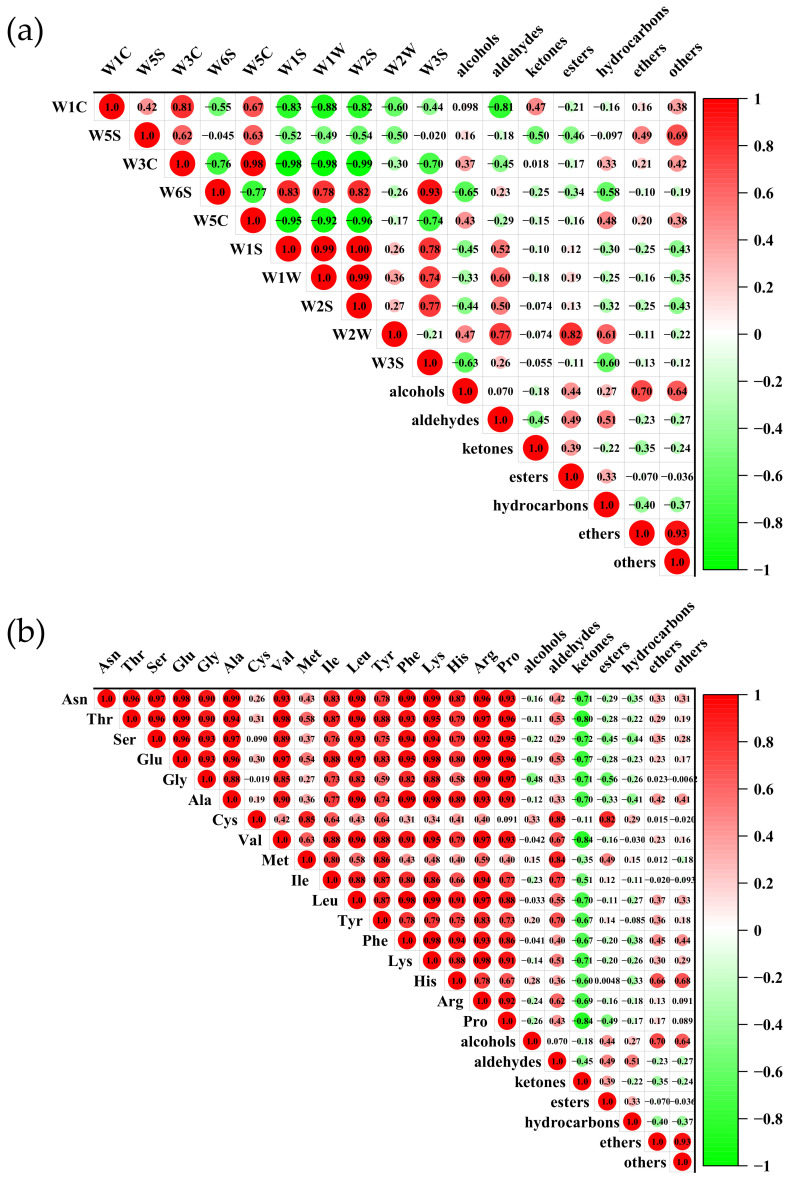
Correlations between (**a**) electronic nose and volatile flavor components and (**b**) free amino acids and volatile flavor components.

**Table 1 foods-11-02594-t001:** Comparison of physicochemical parameters in different parts of JPGC and CGC.

Samples	Content (g/100g)					
	JA	CA	JB	CB	JT	CT
Crude protein	20.14 ± 0.38 ^b^	17.59 ± 0.23 ^d^	21.55 ± 0.57 ^a^	18.50 ± 0.46 ^c^	20.04 ± 0.33 ^b^	16.86 ± 0.28 ^d^
Crude fat	4.84 ± 0.13 ^b^	6.13 ± 0.12 ^a^	4.70 ± 0.19 ^b^	5.94 ± 0.21 ^a^	4.46 ± 0.15 ^b^	5.98 ± 0.12 ^a^
Moisture	73.92 ± 4.24 ^a^	75.79 ± 4.78 ^a^	72.95 ± 4.89 ^a^	75.29 ± 1.99 ^a^	74.98 ± 2.32 ^a^	76.36 ± 5.92 ^a^
Ash	1.18 ± 0.01 ^ab^	1.05 ± 0.02 ^d^	1.22 ± 0.01 ^a^	1.09 ± 0.01 ^c^	1.16 ± 0.02 ^b^	1.05 ± 0.01 ^d^

Different superscript letters in the same row indicate significant differences (*p* < 0.05). JPGC: grass carp from Jingpo Lake; CGC: commercial grass carp; JA: abdomen of JPGC; JB: back of JPGC; JT: tail of JPGC; CA: abdomen of CGC; CB: back of CGC; CT: tail of CGC.

**Table 2 foods-11-02594-t002:** Contents (μg/kg) of volatile compounds in different parts of JPGC and CGC.

No	Volatile Compounds	Content (μg kg^−1^)					
		JA	CA	JB	CB	JT	CT
Alcohols							
1	1-Octanol	–	–	1.87 ± 0.12 ^a^	1.20 ± 0.02 ^b^	–	–
2	1,3,5-Benzetriol	1.05 ± 0.03 ^a^	–	0.87 ± 0.01 ^a^	–	–	–
3	11-Bromo-1-undecanol	–	–	1.78 ± 0.01 ^a^	–	–	–
4	Palmidrol	–	–	2.66 ± 0.23 ^a^	–	–	–
5	Sorbitol	–	–	1.49 ± 0.01 ^a^	–	–	–
6	1-Tetradecanol	–	–	2.64 ± 0.21 ^b^	–	4.99 ± 0.13 ^a^	–
7	Hexaethylene glycol	–	–	2.26 ± 0.19 ^a^	–	–	–
8	Isoamyl alcohol	–	–	1.06 ± 0.01 ^a^	–	–	–
9	2,5,8,11,14-Pentaoxahexadecan-16-ol	–	–	1.38 ± 0.08 ^a^	–	–	–
10	Glycerol, 3TBDMS derivative	–	–	1.52 ± 0.12 ^a^	–	–	–
11	3,4-Dihydroxyphenylglycol, 4TMS derivative	–	–	1.53 ± 0.18 ^a^	–	–	–
12	Tetraethylene glycol	–	–	1.50 ± 0.10 ^a^	–	–	–
13	2-[2-[2-[2-[2-[2-[2-(2-Hydroxyethoxy)ethoxy]ethoxy]ethoxy]ethoxy]ethoxy]ethoxy]ethanol	–	–	0.92 ± 0.02 ^a^	–	–	–
14	4-Octanol, 4,7-dimethyl-	–	–	2.64 ± 0.20 ^a^	–	–	–
15	1,5-Anhydro-2-O-acetyl-3,4,6-tri-O-methyl-D-glucitol	–	–	0.94 ± 0.01 ^a^	–	–	–
16	(R)-2,4-Dihydroxy-N-(3-hydroxypropyl)-3,3-dimethylbutyramide	–	–	1.07 ± 0.01 ^a^	–	–	–
17	1-nonanol	0.81 ± 0.02 ^b^	1.45 ± 0.02 ^a^	–	1.39 ± 0.02 ^a^	–	–
18	1-Hexanol	19.88 ± 0.69 ^a^	21.11 ± 0.87 ^a^	20.71 ± 0.59 ^a^	22.93 ± 0.54 ^a^	17.59 ± 0.60 ^b^	17.64 ± 0.61 ^b^
19	4-Methylmannitol	–	–	–	–	2.68 ± 0.02 ^a^	–
20	1-Butanol,3-(1-ethoxyethoxy)-4,4,4-trifluoro-	–	–	–	–	2.34 ± 0.01 ^a^	–
21	Triethylene glycol monododecyl ether	–	–	–	1.45 ± 0.02 ^b^	2.68 ± 0.02 ^a^	–
22	Glycerol, 1,2-di(TMS)-	–	–	–	4.00 ± 0.03 ^a^	–	–
23	3-Methyl-5-methoxy-1-pentanol	–	–	–	1.34 ± 0.01 ^a^	–	–
24	3-Methoxy-hexane-1,6-diol	–	–	–	1.27 ± 0.02 ^a^	–	–
25	3,7,11,15-Tetramethyl-2-hexadecen-1-ol	–	5.62 ± 0.26 ^b^	6.17 ± 0.04 ^a^	–	–	–
26	1,2,4-Cyclopentanetriol	–	1.32 ± 0.02 ^a^	–	–	–	–
27	2-Octyldecanol	–	–	–	–	–	0.44 ± 0.01 ^a^
28	Nonaethylene glycol	–	–	–	–	–	23.67 ± 0.18 ^a^
29	Fucoxanthin	–	–	–	–	2.95 ± 0.12 ^a^	–
Total alcohols		21.74 ± 0.74 ^f^	29.50 ± 1.17 ^e^	52.47 ± 2.14 ^a^	33.58 ± 0.66 ^c^	33.23 ± 0.90 ^c^	41.75 ± 0.80 ^b^
Aldehydes							
30	2,5-Dihydroxybenzaldehyde	3.77 ± 0.02 ^c^	4.34 ± 0.08 ^b^	3.00 ± 0.10 ^e^	4.16 ± 0.02 ^b^	6.61 ± 0.07 ^a^	4.30 ± 0.06 ^b^
31	1-nonanal	3.52 ± 0.10 ^a^	1.07 ± 0.03 ^d^	3.59 ± 0.03 ^a^	1.13 ± 0.01 ^d^	3.06 ± 0.11 ^b^	2.58 ± 0.05 ^c^
32	3-Methylbutanal	6.97 ± 0.05 ^a^	4.16 ± 0.24 ^d^	5.81 ± 0.34 ^c^	3.91 ± 0.32 ^e^	6.42 ± 0.56 ^b^	3.93 ± 0.16 ^e^
33	Hexanal	–	–	–	–	2.37 ± 0.02 ^a^	–
34	Vanillin	4.38 ± 0.10 ^a^	–	3.43 ± 0.12 ^b^	–	–	–
35	3-Hydroxy-4-methoxybenzaldehyde	–	–	–	2.72 ± 0.05 ^b^	4.25 ± 0.16 ^a^	–
36	2-dodecenal	–	–	–	–	1.91 ± 0.03 ^b^	3.98 ± 0.09 ^a^
Total aldehydes		18.64 ± 0.29 ^b^	9.57 ± 0.35 ^e^	15.83 ± 0.59 ^c^	11.92 ± 0.40 ^d^	24.62 ± 0.95 ^a^	14.79 ± 0.36 ^c^
Ketones							
37	Xanthoxylin	–	–	–	8.65 ± 0.17 ^a^	–	–
38	5-Decanone	–	–	–	–	–	1.34 ± 0.02 ^a^
39	5,5-dichloro-4-Spirohexanone	–	1.82 ± 0.08 ^a^	–	–	–	–
Total ketones		0	1.82 ± 0.08 ^b^	0	8.65 ± 0.17 ^a^	0	1.34 ± 0.02 ^c^
Esters							
40	Butyric acid pentadecyl ester	–	–	–	2.62 ± 0.11 ^a^	–	–
41	Propanoic acid, dimethyl (ethenyl)silyl ester	–	–	–	–	–	1.95 ± 0.04 ^a^
42	Diglycolic acid, 2-chloro-6-fluorophenyl nonyl ester	–	–	–	–	–	1.86 ± 0.02 ^a^
43	Diethyl sulfate	–	–	0.86 ± 0.02 ^a^	–	–	–
44	Methyl myristoleate	–	–	1.30 ± 0.03 ^a^	–	–	–
45	Isopropyl palmitate	0.89 ± 0.02 ^b^	–	2.29 ± 0.14 ^a^	–	–	–
46	Triethyl borate	–	–	1.11 ± 0.01 ^a^	–	–	–
47	Glycerol monostearate	–	–	1.58 ± 0.08 ^a^	–	–	–
48	di(Butoxyethyl)adipate	–	–	2.04 ± 0.10 ^a^	–	–	–
49	Methyl 2-hydroxystearate	–	–	0.89 ± 0.02 ^a^	–	–	–
50	Dimethyl dl-malate	–	–	0.84 ± 0.01 ^a^	–	–	–
51	Methyl acetylglycinate	–	–	0.80 ± 0.03 ^a^	–	–	–
52	Isobutyl 3-hydroxy-2-methylenebutanoate	–	–	0.85 ± 0.01 ^a^	–	–	–
53	Hexanoic acid, cyclohexyl ester	–	–	0.86 ± 0.02 ^a^	–	–	–
54	Propanoic acid, 2-methyl-, octyl ester	–	–	1.06 ± 0.01 ^a^	–	–	–
55	Hexacosanoic acid, methyl ester	–	–	1.60 ± 0.09 ^a^	–	–	–
56	Heptanoic acid, octyl ester	–	–	0.94 ± 0.03 ^a^	–	–	–
57	Methyl benzoate	3.19 ± 0.13 ^e^	–	6.24 ± 0.29 ^b^	5.21 ± 0.10 ^c^	7.66 ± 0.32 ^a^	6.12 ± 0.21 ^b^
58	Propanoic acid, 3-ethoxy-, ethyl ester	–	–	1.86 ± 0.08 ^a^	–	–	–
59	Heptanoic acid, propyl ester	–	–	0.87 ± 0.05 ^a^	–	–	–
60	ethyl-2-methylbutanoate	–	–	6.01 ± 0.62 ^a^	–	–	–
61	ENT-337	–	–	0.89 ± 0.02 ^a^	–	–	–
62	L-Citrulline, *N*,*N*’-bis(dimethylaminomethylene)-, methyl ester	–	–	1.88 ± 0.07 ^a^	–	–	–
63	2-Isoxazolidinecarboxylic acid, ethyl ester	–	–	0.79 ± 0.02 ^a^	–	–	–
64	Silicic acid, diethyl bis(trimethylsilyl) ester	3.62 ± 0.22 ^b^	–	–	4.18 ± 0.31 ^a^	–	–
65	Glycylglycine ethyl ester	1.11 ± 0.02 ^a^	–	–	–	–	–
66	Carbamodithioic acid, diethyl-, methyl ester	3.67 ± 0.31 ^b^	–	–	9.42 ± 0.49 ^a^	–	–
67	Imidodicarbonic acid, diethyl ester	0.95 ± 0.05 ^a^	–	–	–	–	–
68	(2-Bromomethyl-[1,3]dioxolan-2-yl)-acetic acid, methyl ester	–	–	–	–	2.42 ± 0.14 ^a^	–
69	Pentyl (3S)-3-hydroxy-5-methoxypentanoate	–	–	–	0.98 ± 0.01 ^c^	3.16 ± 0.16 ^a^	2.45 ± 0.08 ^b^
70	Butanoic acid, 4-ethoxy-, methyl ester	–	–	–	–	3.93 ± 0.28 ^a^	–
71	(E)-3,7,11-Trimethyldodec-2-enoic acid, methyl ester	–	–	–	–	1.80 ± 0.09 ^a^	–
72	Heptadecanoic acid, 9-methyl-, methyl ester	–	–	–	–	4.28 ± 0.44 ^a^	–
73	Pentanoic acid, 2-hydroxy-3-methyl-, methyl ester	–	–	–	–	2.70 ± 0.10 ^a^	–
74	Boric acid (H3BO3), tris(1-methylethyl) ester	–	–	–	–	1.52 ± 0.09 ^a^	–
75	4-Methylmannonic.delta.- lactone	–	–	–	–	1.99 ± 0.11 ^a^	–
76	3-(1-Ethoxy-ethoxy)-2-ethyl-butyric acid, ethyl ester	–	–	–	–	3.31 ± 0.28 ^a^	–
77	Isobutyl 2,5,8,11-tetraoxatridecan-13-yl carbonate	–	–	–	–	2.10 ± 0.12 ^a^	–
78	3-Deoxy-d-mannoic lactone	–	–	–	0.98 ± 0.02 ^a^	–	–
79	Propanoic acid, 2-methyl-, decyl ester	–	–	–	1.76 ± 0.15 ^a^	–	–
80	l-(+)-Ascorbic acid 2,6-dihexadecanoate	–	–	–	1.28 ± 0.04 ^a^	–	–
81	Trimethylene borate	–	–	–	3.17 ± 0.36 ^a^	–	–
82	Methyl octa-O-methyllaminaribionate	–	–	–	1.10 ± 0.09 ^a^	–	–
83	Methyl-(aminosulfanyl)formate	–	–	–	1.14 ± 0.04 ^a^	–	–
84	Propyl (3S)-3-hydroxy-5-methoxypentanoate	–	–	–	2.21 ± 0.20 ^a^	–	–
85	Ethanol, 2-[2-(2-methoxyethoxy)ethoxy]-, acetate	–	–	–	1.30 ± 0.07 ^a^	–	–
86	Tetraethyleneglycol monomethylether	–	–	–	0.99 ± 0.03 ^a^	–	–
87	Butanoic acid, nonyl ester	–	1.83 ± 0.24 ^a^	–	–	–	–
88	Methyl 3-hydroxytetradecanoate	–	–	–	–	–	1.37 ± 0.11 ^a^
89	Octaethylene glycol monododecyl ether	–	–	–	–	–	2.07 ± 0.09 ^a^
90	Nonanoic acid, 9-oxo-, ethyl ester	–	–	–	–	–	1.60 ± 0.13 ^a^
91	Methyl 5-methoxy-3-oxovalerate	–	–	–	–	–	1.28 ± 0.07 ^a^
92	7-Hexadecenoic acid, methyl ester, (Z)-	–	–	–	–	–	9.15 ± 0.83 ^a^
Total esters		13.43 ± 0.75 ^d^	1.83 ± 0.24 ^e^	35.56 ± 1.82 ^b^	36.34 ± 2.02 ^b^	42.27 ± 2.13 ^a^	27.85 ± 1.61 ^c^
Hydrocarbons							
93	Hexane, 3-methyl-	–	–	1.44 ± 0.08 ^a^	–	–	–
94	Trichloromethane	–	–	1.20 ± 0.03 ^a^	1.23 ± 0.04 ^a^	–	–
95	Heneicosane	7.34 ± 0.52 ^d^	2.76 ± 0.14 ^e^	2.32 ± 0.12 ^f^	9.91 ± 0.43 ^c^	28.41 ± 0.98 ^b^	33.37 ± 2.09 ^a^
96	Pentatriacontane	–	–	3.05 ± 0.24 ^a^	–	–	–
97	Dodecane, 2-methyl-	–	–	1.64 ± 0.10 ^a^	–	–	–
98	Propane, 1,1’-[ethylidenebis(oxy)]bis-	–	–	1.05 ± 0.06 ^a^	–	–	–
99	trans-Calamenene	–	–	0.93 ± 0.06 ^a^	–	–	–
100	Heptadecane	1.92 ± 0.10 ^b^	–	–	–	5.6 ± 0.30 ^a^	–
101	cis-Calamenene	1.55 ± 0.19 ^b^	–	–	–	3.26 ± 0.24 ^a^	–
102	Oxetane, 2-propyl-	–	–	–	–	–	1.31 ± 0.09 ^a^
103	Heptadecane, 7-methyl-	–	–	–	–	–	18.73 ± 2.05 ^a^
104	1,4,7-Triazacyclononane, 1-benzoyl-	–	–	–	–	–	2.69 ± 0.08 ^a^
105	1-Tetracosene	–	–	–	–	–	3.99 ± 0.23 ^a^
106	1-Eicosyne	–	–	–	–		2.00 ± 0.10 ^a^
Total hydrocarbons		10.81 ± 0.81 ^e^	2.76 ± 0.14 ^f^	11.63 ± 0.79 ^c^	11.14 ± 0.47 ^d^	37.27 ± 1.52 ^b^	62.09 ± 4.64 ^a^
Ethers							
107	Estragole	–	–	4.93 ± 0.65 ^a^	–	–	–
108	2-(2-Methoxyethoxy)ethanol	–	–	1.07 ± 0.02 ^a^	–	–	–
109	2-[2-(hexyloxy)ethoxy]-ethanol	–	–	0.93 ± 0.05 ^a^	–	–	–
110	Tetraethylene glycol diethyl ether	–	–	1.04 ± 0.10 ^a^	–	–	–
111	Ethanol, 2-(2-ethoxyethoxy)-	–	–	0.78 ± 0.02 ^a^	–	–	–
112	Undecaethylene glycol monomethyl ether	1.16 ± 0.10 ^a^	–	–	–	–	–
113	Eicosyl methyl ether	–	2.25 ± 0.16 ^a^	–	–	–	–
114	Anethole	3.12 ± 0.16 ^c^	2.90 ± 0.10 ^c^	4.28 ± 0.29 ^b^	2.63 ± 0.11 ^e^	6.43 ± 0.46 ^a^	2.39 ± 0.08 ^d^
Total ethesrs		4.28 ± 0.26 ^d^	5.15 ± 0.26 ^c^	13.03 ± 1.13 ^a^	2.63 ± 0.11 ^e^	6.43 ± 0.46 ^b^	2.39 ± 0.08 ^f^
Others							
115	3-Phenylindole	–	–	1.76 ± 0.08 ^a^	–	–	–
116	1,2-Bis(trimethylsilyl)benzene	–	–	0.91 ± 0.03 ^a^	–	–	–
117	Butylated Hydroxytoluene	2.61 ± 0.26 ^b^	–	3.59 ± 0.15 ^a^	–	–	–
118	Azulene	–	1.38 ± 0.08 ^a^	–	–	–	–
119	Naphthalene	–	–	–	1.45 ± 0.08 ^a^	–	1.36 ± 0.06 ^a^
Total others		5.73 ± 0.48 ^c^	4.28 ± 0.38 ^d^	10.54 ± 0.55 ^a^	4.08 ± 0.19 ^e^	6.43 ± 0.46 ^b^	3.75 ± 0.16 ^f^

(–): volatile flavor compounds not detected. Different lowercase letters in the same row indicate a significant difference (*p* < 0.05). JPGC: grass carp from Jingpo Lake; CGC: commercial grass carp; JA: abdomen of JPGC; JB: back of JPGC; JT: tail of JPGC; CA: abdomen of CGC; CB: back of CGC; CT: tail of CGC. MS: identification based on the NIST mass spectrometry database.

**Table 3 foods-11-02594-t003:** Comparison of amino acids in different parts of JPGC and CGC.

Amino Acid Species	Taste Attribute	Content (g Amino Acid/100 g Protein)					
2222		JA	CA	JB	CB	JT	CT
Aspartic acid ^▲^ (Asp)	Umami/sour (+)	5.96 ± 0.17 ^a^	5.46 ± 0.27 ^c^	5.69 ± 0.13 ^b^	5.15 ± 0.21 ^d^	5.70 ± 0.16 ^b^	5.18 ± 0.23 ^d^
Threonine ^▲^ (Thr)	Sweet (+)	2.01 ± 0.08 ^a^	1.92 ± 0.03 ^ab^	1.96 ± 0.09 ^ab^	1.80 ± 0.07 ^b^	2.00 ± 0.04 ^a^	1.83 ± 0.17 ^b^
Serine ^▲^ (Ser)	Sweet (+)	2.11 ± 0.02 ^a^	2.03 ± 0.04 ^ab^	2.04 ± 0.04 ^ab^	1.91 ± 0.06 ^b^	2.05 ± 0.03 ^ab^	1.92 ± 0.02 ^b^
Glutamic acid ^▲^ (Glu)	Umami/sour (+)	7.94 ± 0.06 ^a^	7.32 ± 0.11 ^d^	7.53 ± 0.12 ^c^	6.77 ± 0.19 ^f^	7.78 ± 0.08 ^b^	6.94 ± 0.18 ^e^
Glycine ^▲^ (Gly)	Sweet (+)	3.20 ± 0.01 ^a^	2.95 ± 0.08 ^b^	2.84 ± 0.04 ^bc^	2.62 ± 0.05 ^d^	2.99 ± 0.01 ^b^	2.73 ± 0.01 ^cd^
Alanine ^▲^ (Ala)	Sweet (+)	3.66 ± 0.16 ^a^	3.37 ± 0.16 ^b^	3.53 ± 0.13 ^ab^	3.17 ± 0.13 ^c^	3.47 ± 0.15 ^b^	3.18 ± 0.12 ^c^
Cysteine (Cys)	Bitter/sweet/sulfur (-)	0.15 ± 0.02 ^a^	0.10 ± 0.01 ^a^	0.18 ± 0.02 ^a^	0.17 ± 0.03 ^a^	0.23 ± 0.01 ^a^	0.15 ± 0.02 ^a^
Valine ^▼^ (Val)	Sweet/bitter (-)	1.57 ± 0.13 ^a^	1.48 ± 0.10 ^a^	1.53 ± 0.07 ^a^	1.42 ± 0.13 ^a^	1.57 ± 0.13 ^a^	1.47 ± 0.06 ^a^
Methionine ^▼^ (Met)	Bitter/sweet/sulfur (-)	1.48 ± 0.03 ^b^	1.46 ± 0.09 ^b^	1.55 ± 0.07 ^b^	1.47 ± 0.16 ^b^	1.74 ± 0.10 ^a^	1.44 ± 0.14 ^b^
Isoleucine ^▼^ (Ile)	Bitter (-)	1.32 ± 0.04 ^a^	1.27 ± 0.12 ^a^	1.29 ± 0.07 ^a^	1.27 ± 0.11 ^a^	1.33 ± 0.15 ^a^	1.26 ± 0.14 ^a^
Leucine ^▼^ (Leu)	Bitter (-)	4.11 ± 0.15 ^a^	3.76 ± 0.17 ^b^	4.01 ± 0.17 ^a^	3.62 ± 017 ^b^	4.04 ± 0.14 ^a^	3.63 ± 0.19 ^b^
Tyrosine ^▼^ (Tyr)	Bitter (-)	1.96 ± 0.08 ^a^	1.93 ± 0.04 ^a^	2.00 ± 0.02 ^a^	1.88 ± 0.04 ^a^	2.04 ± 0.05 ^a^	1.89 ± 0.10 ^a^
Phenylalanine ^▼^ (Phe)	Bitter (-)	2.13 ± 0.18 ^a^	1.93 ± 0.17 ^bc^	2.07 ± 0.13 ^ab^	1.84 ± 0.12 ^c^	2.03 ± 0.12 ^ab^	1.83 ± 0.11 ^c^
Lysine ^▼^ (Lys)	Sweet/bitter (-)	4.87 ± 0.15 ^a^	4.38 ± 0.13 ^c^	4.63 ± 0.16 ^b^	4.18 ± 0.18 ^d^	4.68 ± 0.14 ^b^	4.23 ± 0.16 ^cd^
Histidine ^▼^ (His)	Bitter (-)	1.57 ± 0.02 ^a^	1.33 ± 0.04 ^bc^	1.63 ± 0.08 ^a^	1.31 ± 0.03 ^bc^	1.47 ± 0.09 ^ab^	1.29 ± 0.07 ^c^
Arginine ^▼^ (Arg)	Sweet/bitter (-)	2.96 ± 0.01 ^a^	2.75 ± 0.08 ^b^	2.82 ± 0.06 ^ab^	2.66 ± 0.08 ^b^	2.90 ± 0.11 ^a^	2.67 ± 0.05 ^b^
Proline ^▲^ (Pro)	Sweet/bitter (+)	2.09 ± 0.03 ^a^	1.97 ± 0.03 ^ab^	1.95 ± 0.03 ^ab^	1.73 ± 0.04 ^c^	2.03 ± 0.05 ^a^	1.85 ± 0.03 ^bc^
Total		49.09 ± 2.54 ^a^	45.42 ± 1.78 ^bc^	47.24 ± 1.20 ^b^	42.96 ± 0.94 ^c^	48.05 ± 2.61 ^ab^	43.46 ± 1.65 ^c^

^▲^: fresh sweet amino acids; ^▼^: bitter amino acids; (+): pleasant taste; (-): unpleasant taste. Different superscript letters in the same row indicate significant differences (*p* < 0.05). JPGC: grass carp from Jingpo Lake; CGC: commercial grass carp; JA: abdomen of JPGC; JB: back of JPGC; JT: tail of JPGC; CA: abdomen of CGC; CB: back of CGC; CT: tail of CGC.

**Table 4 foods-11-02594-t004:** Comparison of ATP-associated compounds and *K*-values in different parts of JPGC and CGC.

ATP-Associated Compounds	Content (mg/kg)					
	JA	CA	JB	CB	JT	CT
ATP	156.39 ± 6.86 ^b^	151.62 ± 5.12 ^c^	171.55 ± 6.51 ^a^	165.04 ± 6.43 ^a^	151.61 ± 5.27 ^c^	148.07 ± 6.11 ^d^
ADP	272.65 ± 10.86 ^b^	231.94 ± 11.23 ^d^	282.26 ± 10.22 ^a^	248.25 ± 11.36 ^c^	218.52 ± 10.13 ^e^	218.21 ± 11.38 ^e^
AMP	84.39 ± 5.92 ^b^	68.66 ± 5.90 ^d^	87.91 ± 5.91 ^a^	60.78 ± 5.90 ^e^	81.19 ± 4.63 ^c^	66.75 ± 6.58 ^d^
IMP	1951.19 ± 36.34 ^d^	2009.11 ± 43.91 ^c^	2258.22 ± 40.46 ^a^	2243.75 ± 43.26 ^a^	2113.24 ± 37.20 ^b^	2101.44 ± 42.98 ^b^
HxR	202.29 ± 12.03 ^e^	212.88 ± 11.45 ^d^	264.10 ± 11.49 ^a^	200.21 ± 11.76 ^e^	258.69 ± 11.95 ^b^	231.49 ± 10.35 ^c^
Hx	72.98 ± 4.52 ^a^	62.23 ± 2.54 ^b^	29.82 ± 2.56 ^e^	41.64 ± 3.43 ^d^	39.00 ± 3.81 ^d^	55.69 ± 2.26 ^c^
k	10.05 ± 0.43 ^a^	10.05 ± 0.41 ^a^	9.50 ± 0.52 ^a^	8.17 ± 0.99 ^a^	10.40 ± 0.38 ^a^	10.18 ± 0.87 ^a^

Different superscript letters in the same row indicate significant differences (*p* < 0.05). JPGC: grass carp from Jingpo Lake; CGC: commercial grass carp; JA: abdomen of JPGC; JB: back of JPGC; JT: tail of JPGC; CA: abdomen of CGC; CB: back of CGC; CT: tail of CGC.

**Table 5 foods-11-02594-t005:** Sensory evaluation in different parts of JPGC and CGC.

Samples	JA	CA	JB	CB	JT	CT
Texture						
Tenderness	4.04 ± 0.26 ^a^	2.82 ± 0.33 ^b^	3.94 ± 0.22 ^a^	2.69 ± 0.36 ^b^	3.36 ± 0.43 ^ab^	3.12 ± 0.38 ^ab^
Springiness	3.91 ± 0.22 ^a^	2.51 ± 0.42 ^b^	3.87 ± 0.41 ^a^	2.59 ± 0.36 ^b^	3.70 ± 0.52 ^a^	2.60 ± 0.43 ^b^
Chewiness	4.23 ± 0.34 ^a^	3.13 ± 0.56 ^ab^	3.92 ± 0.46 ^a^^b^	3.85 ± 0.26 ^a^^b^	2.83 ± 0.20 ^a^^b^	2.71 ± 0.55 ^b^
Odor						
Fishy	2.38 ± 0.31 ^b^	3.47 ± 0.29 ^a^	2.17 ± 0.51 ^b^	3.44 ± 0.49 ^a^	2.18 ± 0.21 ^b^	3.60 ± 0.28 ^a^
Earthy	1.20 ± 0.32 ^b^	3.11 ± 0.34 ^a^	1.18 ± 0.18 ^b^	3.88 ± 0.41 ^a^	1.54 ± 0.40 ^b^	2.99 ± 0.30 ^a^
Taste						
Fishy	2.87 ± 0.28 ^ab^	3.61 ± 0.37 ^a^	2.75 ± 0.25 ^b^	3.65 ± 0.18 ^a^	2.80 ± 0.29 ^b^	3.18 ± 0.33 ^ab^
Earthy	1.10 ± 0.37 ^b^	3.24 ± 0.40 ^a^	1.45 ± 0.49 ^b^	3.26 ± 0.54 ^a^	1.68 ± 0.12 ^b^	3.29 ± 0.29 ^a^

Different superscript letters in the same row indicate significant differences (*p* < 0.05). JPGC: grass carp from Jingpo Lake; CGC: commercial grass carp; JA: abdomen of JPGC; JB: back of JPGC; JT: tail of JPGC; CA: abdomen of CGC; CB: back of CGC; CT: tail of CGC.

## Data Availability

The data are contained within the article.

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
