# Peer review of "Flavor Differences of Edible Parts of Grass Carp between Jingpo Lake and Commercial Market"

_foods, 2022, doi:10.3390/foods11172594_

Round 1
Reviewer 1 Report
Flavor differences among three individual parts (abdomen, 15 back, and tail) of Jingpo Lake grass carp (JPGC) and commercial grass carp (CGC) were evaluated.
At the end of the introduction section it is stated that in this study, the abdominal, back, and tail muscles of Jingpo Lake grass carp (JPGC) and commercial grass carp (CGC) were by analysed by E-nose, E-tongue, and SPME-GC-MS (volatile compounds -VOCs) to distinguish their odor and taste. Moreover, it is explained that the VOCs were qualitatively and quantitatively identified by GC-MS, and the types and concentrations of FAAs and ATP-related products were determined to identify the characteristic VOCs and taste substances in different parts of JPGC. These results can provide a theoretical basis for the utilization of Jingpo Lake regional grass carp varieties. However, the goals of this study are not clearly presented.
Section 2.2. Sample preparation
Lines 97 and 98: “…each cut of meat was individually vacuum-packaged and stored at −80 ºC until use (within 7 days)…” Sample preservation at -80ºC leads to lose of volatile compounds, it is true that this will occur in all samples, but the volatile profile will not correspond to real. Please comment.
Section 2.3. Volatile compounds analysis
Lines 103-104: “The volatile compounds were extracted from the samples using headspace solid-103 phase microextraction (HS-SPME)…” Details concerning fiber composition are lacking.
Section 3.1. Volatile compounds
Table 1 title is Contents (μg/kg) of volatile compounds… Moreover Lines 159, 163, 165, mention: “The total alcohol content… with the main volatile compounds being alcohols, which were more abundant in the dorsum than in the abdomen… small amount” however, volatile compounds quantification is not described in the experimental section. How was it done the quantification of such great number of volatile compounds?
Line 228-229: “Although 15 hydrocarbons were identified, they had little effect on the overall flavor of grass carp because of their high odor threshold values.” The odor threshold values were not evaluated in this study, probably a reference is missing?
Section 3.2 E-Nose
Lines 253 and 254: “The PCA results of the six grass carp samples are presented in Figure 3(a).” It is not clear which were the variables of this PCA analyses, please explain. The same question applies for Fig 4(a).
4. Conclusions
“Notable differences were observed in the numbers and contents of volatile com-382 pounds among three individual parts (abdomen, back, and tail) of JPGC and CGC.” Why this is relevant?
The relevance and the novelty of this study must be highlighted. Authors state that “This study can provide useful information for elucidating the flavor characteristics of edible parts of JPGC.” Sensory studies could be more appropriate.
Reviewer 2 Report
The manuscript submitted by Chen et al. deals with the evaluation of flavor differences between different portions of Jingpo Lake grass carp and commercial grass carp, by using three different approaches.
The topic concerning the comparison between E-nose, E-tongue, and SPME-GC-MS is not so original, since in the literature there are different studies in which these three methodologies are compared; the major point of interest can be however found in the matrix under analysis (Jingpo Lake grass carp vs commercial grass carp).
Major points:
Based on the comparative picture (Figure 1), it seems that the two types of carp have a different morphology. I wonder if on a genetic level these fish have big differences that could possibly justify a different metabolism. This, from my point of view, would represent a serious failure for the experimental design of the study since the observed differences could derive from variables other than the simple origin of the sample (lake vs commercial). In addition to this, it must be considered that the two types of fish are subjected to a different motor activity that will certainly have influenced muscle development and, consequently, the quality and quantity of intramuscular fat which, as known, carries most of the flavor compounds.
In addition, I believe there is also the ethical issue for carp not obtained from the supermarket. The authors do not explain how they got these fish and how the slaughter happened. However, I believe that approval by an ethics committee is mandatory.
The presentation and discussion of the results is quite clear and straightforward, however these sections of the paper could be influenced by the poor solidity of the experimental design.
Minor points:
- The title must be modified; the grammar is not correct.
- In the introduction must be reported the hypothesis on which the study is based. What is reported in lines 70-71 is not enough to justify the experimentation.
- Since the study is also based on the comparison between different techniques, I think it is useful to add more details in the M&M section, without just mentioning other references.
Reviewer 3 Report
The manuscript by Chen et al. addresses an interesting topic of flavor analysis of fish samples.
major concerns
the content of each VOCs cannot be calculated without standards, which are not listed in 2.1 section of Materials.
Conclusions part is not accurate with title. Authors concluded that there is a significant difference in different parts of fish, however the title suggests that the study of difference between commercial and Jingpo lake samples will be the topic. The question “which samples commercial and Jingpo lake is more flavorable”.
Moreover, is there any sensorial/consumers study due to the topic of this species of fish?
some minor concerns
English language requires editing, ideally by a native speaker. At times, grammatical errors and the tendency to write long sentences turn the text confusing and difficult to follow.
L: 49 I suggest to edit expression “volatile odor” to “volatiles” or “odors”. Also, “odor-active” is not appropriate form.
The full term of amino acids should be added in the figure legend.Table 1. delete comma in the units also in some lines the statistics is missed.
Table 2 and 3. change 'comtent' to 'content'.
Figure 3 and 4. The description of samples should be added.
Round 2
Reviewer 2 Report
The paper has been substantially improved.
I only have a concern related to the ethical issue; authors stated that "All methods used in this study complied with the Chinese National Guidelines for the use and care of laboratory animals. The animal experiment protocol was approved by the Science and Technology Ethics Committee of Heilongjiang Bayi Agricultural University", however no reference was made to a specific document.
I think authors should mention a specific document and provide it to the Editor
Reviewer 3 Report
The authors made a significant improvement in the manuscript. I have no more comments.
Author Response
Sincerely, we appreciate your suggestions to greatly improve our paper.